# Effect of Air-Abraded Versus Laser-Fused Fluorapatite Glass-Ceramics on Shear Bond Strength of Repair Materials to Zirconia

**DOI:** 10.3390/ma14061468

**Published:** 2021-03-17

**Authors:** Alaaeldin Elraggal, Nikolaos Silikas

**Affiliations:** 1Division of Dentistry, School of Medical Sciences, Faculty of Biology, Medicine and Health, The University of Manchester, Manchester M13 9PL, UK; Nikolaos.silikas@manchester.ac.uk; 2Conservative Dentistry Department, Faculty of Dentistry, Alexandria University, Alexandria 21568, Egypt

**Keywords:** zirconia, shear bond strength, fluorapatite glass-ceramics, XRD, MDP, feldspathic porcelain, lithium disilicate

## Abstract

Zirconia repair could be a feasible alternative option to total replacement in fractured zirconia-based restorations. Maximising the bond strength by enriching zirconia with fluorapatite glass-ceramics (FGC) powder has been addressed and compared to other surface treatments. Besides resin composite, other repair materials have been proposed and compared. Zirconia blocks received different surface treatments (A—sandblasting with tribochemical silica-coated alumina (CoJet). B—sandblasting with FGC powder (FGC), C—fluorapatite glass-ceramic coat+ neodymium-doped yttrium aluminum garnet laser irradiation (FGC + Nd: YAG), and D—no surface treatment). The surface roughness, topography, and crystallinity were investigated by a profilometer, scanning electron microscopy (SEM), and X-ray diffraction (XRD) analyses, respectively. For each surface treatment, three repair materials (feldspathic porcelain, lithium disilicate, and resin composite) were bonded to zirconia with 10, Methacryloyloxydecyl dihydrogen phosphate (MDP)–Monobond Plus/ Multilink Automix. Bonded specimens were thermocycled for 10,000 cycles and tested for shear bond strength (SBS) at a speed of 1 mm/min, followed by the analysis of the mode of failure. FGC + Nd: YAG laser group reported the highest surface roughness and monoclinic content compared to CoJet, FGC, and control groups. The highest mean SBS was found in FGC-blasted zirconia, followed by FGC + Nd: YAG laser and CoJet treated groups. However, the lowest SBS was found in control groups regardless of the repair material. Sandblasting zirconia with FGC powder increased SBS of resin to zirconia with lower monoclinic phase transformation compared to FGC + Nd: YAG or CoJet groups.

## 1. Introduction

Zirconia-based restorations (ZBR) have been widely used in dentistry due to their superior mechanical properties and excellent aesthetics [1]. Zirconia has a fracture toughness of 7–19 MPa m^1/2^ and flexural strength of 1200 MPa, therefore it could act as a durable framework for dental restorations [2]. ZBR has an opaque zirconia substructure which is traditionally veneered by tooth-coloured glass-ceramics [3]. Zirconia substructures usually require mechanical surface treatments, such as sandblasting, grinding, or additional sintering cycles before veneered with glass-ceramics [4,5]. Unfortunately, these treatments have been found to induce surface microcracks that could compromise the interface between zirconia substructure and the veneering porcelain leading to possible delamination of the veneering glass-ceramics exposing the underlying zirconia [6]. Heintze and Rousson classified the severity of fracture of the veneering porcelain with the recommended repair approaches as Grade 1, referring to minor porcelain fractures that could be treated by intraoral finishing and polishing; Grade 2, indicating moderate fractures to be treated by resin composite; and Grade 3, which is given to extensive fractures that need total replacement of the prosthesis [7]. The complete replacement of the prosthesis is not a feasible option in the treatment plan. In this case, the clinician will spend much time removing a resin-bonded restoration with a risk to endanger the abutment. Furthermore, the patient will pay extra cost to remove and reconstruct the fractured prosthesis [8].

Repairing fractured ZBR would be an attractive option for a cost-effective and time-saving approach instead of total replacement [1,9,10]. The fractured segment of ZBR has been routinely replaced by direct resin composites [1]. In the current study, feldspathic and lithium disilicate glass-ceramics were used as repair materials to zirconia in addition to resin composites to provide the clinicians with more comprehensive options of repair materials.

Establishing a durable bond between repair material and zirconia is questionable. In glass-ceramics, hydrofluoric acid etching (HF 9%), followed by a silane application, has been the routine mechanical and chemical treatments to modify the surface topography and enhance the bond strength of glass-ceramics to resin composites, respectively. HF attacks the glassy matrix of glass-ceramics and creates micromechanical locks for resin bonding [11]. The absence of the glassy phase in zirconia makes HF etching and silane treatment ineffective [12]. Several surface treatments to zirconia are utilised in order to improve the bond strength of resin composites to zirconia, such as sandblasting, grinding, and laser irradiation [13,14,15,16]. As an attempt to increase the surface content of zirconia with glass, sandblasting zirconia with tribochemical silica-coated alumina particles has been used. In this procedure, alumina particles act as a vector coated by a layer of silica that bombard zirconia creating micromechanical surface pores. Apart from the micromechanical pores created, silica coats dislodge from alumina carriers and embed into the zirconia surface [10]. Only one study has investigated the effect of sandblasting zirconia with alumina-free glass-ceramic powders on bond strength of resin composites to zirconia. Sandblasted zirconia samples with alumina-free glass-ceramic powder showed higher shear bond strength (24 MPa) to resin composite compared to the blasted specimens by tribochemical silica-coated alumina (18.7 MPa) [17].

Fluorapatite glass-ceramics (FGC) has been recently used as a veneering material to zirconia core. It showed to be compatible with zirconia with reported higher shear bond strength than veneering feldspathic glass-ceramics [18]. IPS e.max ZirPress is a fluorapatite glass-ceramic manufactured by Ivoclar Vivadent (AG, Liechtenstein). It is presented in the form of ingots to be pressed over the zirconia core and showed excellent bond strength with the zirconia core of 40.4 MPa [18]. In addition to silica-coated alumina, FGC powder could be an alternative sandblasting approach to change surface topography and increase the surface content of zirconia with glass-ceramic. However, this has not been yet reported in the literature.

Following sandblasting, priming zirconia surface with phosphate monomer 10- methacryloyloxydecyl dihydrogenphosphate (10-MDP) has been considered the next key step for a durable repair of ZBR. 10-MDP is a phosphate monomer with bifunctional molecules that chemically bond to zirconia and resin composite [19,20].

Dental lasers, such as neodymium-doped yttrium aluminium garnet (Nd: YAG), have been widely applied in dentistry for different dental procedures; treatment of hypersensitivity, cavity preparations, and office bleaching, to name a few [15]. Nd: YAG laser has been recently used in photobiomodulation therapy that enhances the wound healing of dental sockets after surgical extraction [21]. Nd: YAG laser can also modify zirconia surface to create microretentive areas for resin bonding [15]. Nd: YAG acts by melting down the zirconia surface to form a blister-like pattern at a power of 2W. A study used Nd: YAG laser to embed glass particles into Ni–Cr alloy by coating the alloy with a slurry of feldspathic porcelain followed by laser irradiation at a power of 8W in an approach termed “silica-lasing” [22].

In the current study, zirconia has been coated with FGC powder followed by Nd: YAG laser irradiation (FGC + Nd: YAG) as an attempt to increase the surface content of zirconia with glass-ceramics. FGC + Nd: YAG is a modified approach to silica-lasing, in which a lab-prepared FGC powder, instead of a slurry of feldspathic porcelain, was used followed by Nd: YAG laser irradiation at a power of 4 W. It is believed that laser power will melt down the FGC particles down to the zirconia surface. However, this hypothetical interaction has never yet been tested and verified.

Therefore, the current study investigated the effect of sandblasting with FGC powders or tribochemical silica-coated alumina and FGC + Nd: YAG approaches on shear bond strength (SBS) of different repair materials (feldspathic and lithium disilicate glass-ceramic, and resin composite) to zirconia. The null hypotheses tested were (i) different surface treatments would not affect SBS of different repair materials to zirconia and (ii) different repair materials would not significantly change the SBS to zirconia.

## 2. Materials and Methods

### 2.1. Preparation of Zirconia Specimens

A total of 20 partially sintered IPS e.max ZirCAD MO (Ivoclar Vivadent, Schaan, Liechtenstein) zirconia blocks (15 mm × 19 mm × 55 mm) were cut into 120 zirconia blocks of 15 mm × 19 mm × 6 mm using a precision diamond saw IsoMet 1000 (Buehler, Germany), as illustrated in Figure 1. The blocks were polished with P1200 silicon carbide paper (Buehler, Germany) and ultrasonically cleaned in distilled water for 10 min to remove milling debris and air-dried. The polished zirconia blocks were sintered using the Programat S1 1600 (Ivoclar Vivadent, Liechtenstein) at 1500 °C for 30 min. Zirconia blocks were then embedded in epoxy resin using mounting moulds Samplkup (Buehler, Germany) with a diameter of (25 mm) and a height of (20 mm).

### 2.2. Preparation of Fluorapatite Glass-Ceramic Powder

A mortar and pestle were used for crushing down the fluorapatite glass-ceramic ingots (IPS e.max ZirPress, Ivoclar Vivadent, Schaan, Liechtenstein) into smaller frits with an average size <5 mm. Each 1 g of the crushed frits was further pulverised using a mixer mill (MM 400, Retsch, Germany) at parameters of 20 Hz for 1 min. The particle size of the obtained powders was measured using a laser diffraction particle size analysis (Matersizer 2000, Malvern Instruments, Worcestershire, UK).

### 2.3. Surface Treatments

Zirconia blocks were randomly assigned to four groups of 30, one of which worked as a control group.

Group A (CoJet): The surface of zirconia specimens were airborne-particle abraded using an intraoral microblasting unit (AquaCare Twin, Velopex, Medivance Instruments Ltd., London, UK) with 30 μm silica-coated alumina particles (CoJet sand, 3M ESPE, Seefeld, Germany) at 3 bar pressure for 15 s at 10 mm distance [23];

Group B (FGC): Zirconia blocks were sandblasted using the ground FGC (IPS e.max ZirPress) powder (53 µm) using sandblasting unit (AquaCare Twin, Velopex, Medivance Instruments Ltd., London, UK) for 15 s under a pressure of 3 bar at a distance of 10 mm;

Group C (FGC+ Nd: YAG): Zirconia surfaces were covered by a custom-made brass metal with central perforation exposing a circular area of zirconia with a diameter of 5 mm. FGC powders were mixed with distilled water and brushed over the exposed zirconia surface using a paintbrush. They were left to dry before Nd: YAG laser irradiation (Fotona d.d, Ljubljana, Slovenia) was applied perpendicular to zirconia at 1mm away, as illustrated in Figure 2. The distance was kept constant among all irradiated specimens by mounting the laser handpiece into a holder device. The parameters were set to be an output power of 4 W, frequency of 20 Hz, short pulse duration of 180 µm, and a total irradiation time of 1 min/specimen;

Group D (Control): As polished.

### 2.4. Zirconia Surface Characterisation

#### 2.4.1. Scanning Electron Microscope (SEM)

One specimen from each group was used for surface topography investigation. Specimens were platinum coated and viewed using a scanning electron microscope (SEM, Zeiss EVO 60, Germany) with an accelerating voltage of 10 kV and a working distance of 8.5–9 mm. Secondary electron images were taken at a magnification of 2000×, followed by energy dispersive spectroscopy (EDS) for elemental analysis. 

#### 2.4.2. Surface Roughness

A total of 40 zirconia specimens (N = 10) were used for surface roughness evaluation by a contact stylus profilometer (Mahr, Marsurf PS10, Germany). Stylus probe with a radius tip of 2 µm and contact force of 0.7 mN travelled a length of 4 mm in the centre of the treated zirconia at speed of 0.5 mm/s, and a cut-off value of 0.08 mm. Five readings, in parallel lines, were measured per specimen. The following roughness parameters were calculated: (i) the arithmetic average of surface profile (Ra), (ii) the maximum average of the highest peak and lowest valleys (Rz), (iii) an average of maximum peaks (Rp), and (iv) the average of deepest valleys (Rv).

#### 2.4.3. X-ray Diffraction (XRD) Analysis

The zirconia phase transformation from tetragonal to monoclinic, as for the blasted and control groups, was investigated using an XRD diffractometer (Lab X XRD-6100, Shimadzu, Japan) with Cu K_α_ X-ray tube head (λ = 1.5416 Å) operated at 40 kv and 30 mA with scanning speed of 12°/min, step size of 0.02° and 1 s/step. XRD peaks were collected at a 2-theta range between 5° to 80°. Phase identification of tetragonal and monoclinic zirconia was conducted using the Shimadzu Xlab software program (6100, Shimadzu, Kyoto, Japan) and compared with the International Centre for Diffraction Data (ICDD) XRD database. The amount and volume fraction of the monoclinic zirconia phase were calculated according to Garvie and Nicholson equation [24] modified by Toraya [25].
(1)$$X_{M}\;=\;\frac{I_{M}\;\left(111^{-}\right)+\;I_{M}\left(111\right)}{I_{M}\;\left(111^{-}\right)+\;I_{M}\left(111\right)+I_{T}\left(101\right)}$$
(2)$$V_{M}\;=\;\frac{1.311\;X_{M}}{1+0.311\;X_{M}}$$
where **X**_M_ is the monoclinic peak intensity ratio in sandblasted zirconia; **V**_M_ is the volumetric fraction of monoclinic phase in sandblasted zirconia. $I_{M}\;\left(111^{-}\right)$ represents the intensity of monoclinic peaks collected at 2θ ≈ 31.2°, $I_{M}\;\left(111\right)$ represents the intensity of monoclinic peaks collected at 2θ ≈ 30°, and $I_{T}\left(101\right)$ represents the intensity of tetragonal peaks collected at 2θ ≈ 28°.

### 2.5. Preparation of Feldspathic Porcelain

A total of 40 cylinders (5 mm diameter and 3 mm thickness) of feldspathic porcelain VITAVM^TM^ 13 (VITA, Zahnfabrik, Bad Sackingen, Germany) were obtained using a custom-made brass mould with inner dimensions of 5 mm diameter and 3 mm in thickness. Feldspathic powder and modelling liquid were mixed using a plastic spatula until the mix was homogenous with a dough-like consistency. The mould was then loaded by the mix and excess liquid was removed by absorbent tissue. A key was then used to dislodge the specimens from the mould. All the specimens were then sintered in a porcelain oven (Programat P310, Ivoclar Vivadent) following the manufacturer’s instructions, as illustrated in Figure 3.

Feldspathic porcelain undergoes ≈20% volumetric sintering shrinkage. For this reason, specimens were returned to the mould and a mixture of porcelain powder and the liquid was added again and then another firing cycle was carried out. Sharp edges were removed using a diamond disc mounted on a low-speed handpiece. Specimens were then finished and polished with P1000 and P1200 silicon carbide papers using grinding machine Metaserv 250 (Buehler Metaserv, Buehler, Germany).

### 2.6. Preparation of Lithium Disilicate Porcelain

Lithium disilicate porcelain (IPS e.max CAD, Ivoclar Vivadent, Schaan, Liechtenstein) is presented in form of rectangular blocks to be milled by a CAD/CAM machine. The rectangular blocks were trimmed into a cylindrical shape with a diameter of 3 mm using a milling machine. A total of 40 cylinders (5 mm in diameter and 3 mm in thickness) of lithium disilicate porcelain were then obtained by sectioning the milled IPS e.max CAD cylinders using Isomet 4000 (Buehler, Germany) at speed of 300 rpm, as illustrated in Figure 4. The obtained cylinders were sintered (Programat P310, Ivoclar Vivadent) following the manufacturer’s recommendations. Sintered cylinders were finished and polished with P1000 and P1200 silicon carbide papers using grinding machine Metaserv 250 (Buehler Metaserv, Buehler, Germany) before ultrasonically cleaned in distilled water and dry-stored.

### 2.7. Bonding Procedure

The surface area of bonding for each zirconia specimen was defined using a pre-perforated adhesive tape covering the zirconia surface exposing only a well-circumscribed round area of 5 mm in diameter. For each surface treatment, repair materials were assigned into three subgroups (n = 40/subgroup) as follows:

Tetric: Adhese Universal + Monobond Plus + Tetric EvoCeram Bulk Fill;

VM 13: Monobond Plus + Multilink Automix + VITAVM ^TM^ 13;

IPS e.max CAD: Monobond Plus + Multilink Automix + IPS e.max CAD.

a-For the Tetric group, zirconia surfaces were rubbed with 10-MDP Monobond Plus for 1 min and air-dried for 5 s before coated with Adhese Universal using microbrush and light-cured for 10 s using a LED curing light (Elipar S10, 3M ESPE) with an output of irradiance of 1200 mW/cm^2^. Custom-made silicone mould with a 5 mm diameter and 3 mm thickness was then placed over the bonding area. The mould was then bulk-filled by Tetric EvoCeram Bulk Fill using a plastic filling instrument before it was light-cured for 20 s.b-For VM 13 and e.max CAD groups, exposed zirconia was treated with 10-MDP Monobond Plus and left for 60 s and then air-dried for 5 s. VM13 and IPS e.max CAD specimens were etched with (HF 9% and HF 5.5%, respectively) for 1 min, water-rinsed, and air-dried before they were treated with Monobond Plus primer for 1 min. Multilink Automix resin cement was extruded over the bonding area using a double syringe through an auto-mixing tip.

For all groups, a mass of 2.25 kg was applied on the top of each resin composite specimen for 5 min to allow the flow of excess resin [26]. Excess cement was then removed at the margins using a microbrush and then light-cured for 20 s. Adhesive tape was then removed and all bonded specimens were subjected to 10,000 thermocycles in water between 5 °C to 55 °C and dwell time of 5 s (Figure 5a). They were then stored for 24 h in a dry place.

### 2.8. Shear Bond Strength (SBS) Test

The bonded resin composite cylinders to zirconia blocks for all groups were mounted into a jig and the shear bond strength was measured using a Universal Testing Machine (Zwick/Roell, Herefordshire, UK) with a knife-edge shearing rod in a load cell capacity of 5000 N at a crosshead speed of 0.5 mm/min (Figure 5b). The maximum shear load at the point of failure was recorded. SBS (σ) was then calculated using the load at failure (F) and the adhesive area (A): σ = F/A (N/mm^2^). 

The mode of failure of debonded specimens was evaluated under SEM (Zeiss EVO 60, Germany). Failure modes were assigned to be [27] as follows:(A)adhesive failure: Failure between the resin composite and zirconia surface;(B)mixed failure: Combined adhesive and cohesive failure in resin composite specimen.

### 2.9. Statistical Analysis

Surface roughness and SBS values were analysed using SPSS for Windows version 22.0 (SPSS Inc., Chicago, IL, USA). The normal distribution of data was explored using Kolmogorov-Smirnov’s test. Data on surface roughness analysis and SBS were found to be normally distributed. Thus, one-way ANOVA was conducted to compare mean values of surface roughness parameters of studied zirconia groups, while two-way ANOVA was used to test SBS, as a dependent variable, on the interaction effect of both surface treatment and repair materials as independent variables. Tukey post hoc test for multiple comparisons was subsequently used to compare the mean SBS between studied groups to detect the statistical difference at a level of significance α = 0.05.

## 3. Results

This section may be divided by subheadings. It should provide a concise and precise description of the experimental results, their interpretation, and the experimental conclusions that can be drawn.

### 3.1. SEM

Images of the scanning electron microscope (2000×) for the tested zirconia surface are presented in Figure 6. Group A (Figure 6a) showed an intensive irregular zirconia surface. However, Group B (Figure 6b) showed fewer irregularities compared to Group A, while Group C (Figure 6c) showed the most extensive irregularities with blister-like deformities on the surface. Group D (Figure 6d) did not reveal any change in the surface topography.

EDS analysis of the percentage mass of silica for differently treated zirconia groups is presented in Figure 7. Group D that received no surface treatments showed no content of silica, while groups (A–C) showed a percentage mass of silica of (1.25 ± 0.13%, 3.42 ± 0.17%, and 6.82 ± 0.21%, respectively.

### 3.2. Surface Roughness

Descriptive statistics of surface roughness parameters and one-way ANOVA results are summarised in Table 1. There was no significant difference in the mean Ra values of control, FGC, and CoJet groups (*p* > 0.05). No significant difference was detected in the Rp value between the control and CoJet groups, while the Rp of the FGC group was statistically higher than the control and CoJet groups. The mean Rv value of FGC was higher than the control group with no significant difference compared to that of CoJet that was significantly higher (*p* < 0.05). Mean Rz values were the lowest in the control group followed by CoJet and FGC groups with a statistical significance found (*p* < 0.05). However, no significant difference was found in Rz between CoJet and FGC. Generally, the FGC + Nd: YAG group recorded the highest statistically significant values of mean Ra, Rp, Rv, and Rz compared to the rest of the groups (*p* < 0.05).

### 3.3. XRD

Diffraction peaks are presented in Figure 8 revealed the presence of monoclinic crystals in all groups, including the control group. The volumetric fraction of monoclinic crystals according to Toraya’s equation was found to be 14.1%, 15.8%, 21.5%, and 26.7% in Groups D, B, A, and C, respectively. The peak of the tetragonal phase (I_T_ -101) was the most accentuated in Group D and most reduced in Group A. Broadening of the tetragonal peak was noticed in treated zirconia surfaces indicating the presence of rhombohedral phase or distorted tetragonal phase.

### 3.4. Shear Bond Strength

Mean SBS values of studied groups and two-way ANOVA statistical comparison were shown in Table 2.

Generally, sandblasting zirconia with FGC powders showed the highest mean SBS, followed by FGC + Nd: YAG approach for different repair materials. CoJet sandblasting reported significantly lower mean SBS (*p* = 0.00) compared to the previous two surface treatments except for zirconia samples repaired with Tetric EvoCeram Bulk Fill. Control groups showed the lowest values for different repair materials, except for IPS e.max CAD- repaired zirconia.

IPS e.max CAD showed the highest mean SBS compared to the other two repair materials (VITAVM ^TM^ 13 and Tetric EvoCeram Bulk Fill) regardless of the surface treatments applied.

The highest mean SBS was found in zirconia specimens treated with FGC sandblasting and repaired with IPS e.max CAD (36.4 ± 2.3 MPa), followed by FGC + Nd: YAG treated group (35 ± 2.2 MPa) with no statistical difference (*p* > 0.05), while CoJet sandblasting and no surface treatment significantly lowered the SBS for the same repair material (25.8 ± 2.40 MPa and 22.7 ± 1.5 MPa, respectively).

For the same surface treatment, there was no statistically significant difference between Tetric EvoCeram Bulk Fill and VITAVM^TM^ 13, except for the CoJet sandblasted zirconia samples; Tetric EvoCeram Bulk Fill reported statistically higher SBS (19.3 ± 0.8 MPa) compared to VITAVM^TM^ 13 (14.2 ± 2.1 MPa).

Two-way ANOVA showed that each of the independent variables (surface treatment and repair materials) or both together had a statistically significant effect (*p* < 0.05) on the shear bond strength values obtained. Repair materials used had a more impact on SBS values than the surface treatment employed.

### 3.5. Modes of Failure

Modes of failure are represented in Figure 9 and Figure 10. Debonded specimens showed predominantly adhesive failure in control groups for different repair materials ranging from (70% in zirconia samples repaired with IPS e.max CAD to 90% in zirconia samples repaired with VITAVM^TM^ 13). However, combined failure in all specimens was found in the FGC group repaired with either of Tetric EvoCeram Bulk Fill or IPS e.max CAD. The same mode of failure was also found in CoJet-sandblasted zirconia repaired with VITAVM^TM^ 13.

In group C, adhesive failure ranged from 20% in the zirconia group repaired with VITAVM^TM^ 13 to 40% in zirconia repaired with IPS e.max CAD.

SEM images (Figure 10) showed a completely adhesive failure in the control group (Figure 10a). Debonded specimens of zirconia, blasted with CoJet sand (Figure 10b), showed mixed adhesive/cohesive failure. Some parts of the resin cement were left behind on the zirconia surface. The exact mode of failure was also found in FGC-blasted zirconia specimens, in addition to a cohesive fracture of the repair material (Figure 10c).

## 4. Discussion

Chipping and delamination of the veneering porcelain are the most common mechanical failures of ZBR [28]. After five years of cementation, a failure rate of 8% [29] and 19% [30] has been reported for single crowns and multiple units fixed partial denture, respectively. A 10-year follow-up clinical trial reported a veneering porcelain chipping rate of 21% of multiple unit ZBR restorations [31]. The repair of the fractured part could be a more convenient approach than total replacement [10]. Resin composites have been routinely used as repair materials for zirconia. However, the bond strength between resin composites and zirconia depends on the mechanical surface treatment to zirconia before the placement of resin composites [1]. This study investigated the effect of sandblasting zirconia with FGC or silica-coated alumina powders in addition to FGC + Nd: YAG laser irradiation on shear bond strength of various repair materials to zirconia. Different repair materials and mechanical surface treatment approach significantly altered the SBS to zirconia. Therefore, both null hypotheses were rejected.

SEM images showed the impact of surface treatments on changing the surface topography of zirconia. The hypothesised interaction between FGC particles and zirconia upon blasting or laser fusion has been confirmed in the current study through the EDS analysis. Progressive increase of silica content from CoJet group (1.25 ± 0.13%), followed by FGC and FGC + Nd: YAG groups (3.42 ± 0.17% and 6.82 ± 0.21%, respectively) was found. These results are consistent with the findings of a previous study [17].

Surface roughness parameters showed the significant impact of FGC + Nd: YAG on zirconia surface compared to other groups. This finding is consistent with previous studies [16,32,33]. However, another study reported that airborne particle abrasion induced significantly higher surface roughness compared to laser-irradiated zirconia specimens [34]. The authors used Er: YAG laser applied directly on zirconia surface without previous coating with glass-ceramics. Further, they used 5 mol%wt Yttria- stabilized tetragonal zirconia polycrystal (Y-TZP), while in our study, 3 mol%wt Y-TZP was used. Different zirconia materials and laser devices might interact differently [34]. Thus, the findings of the current study are applied to 3Y-TZP IPS e.max ZirCAD. Rp and Rv are the surface roughness parameters, representing the maximum peak height and maximum valley depth, respectively [35]. Rp value of the FGC group was significantly higher than its corresponding in control and CoJet groups. The embedding of FGC particles into the zirconia surface in the FGC group might have significantly increased the Rp values compared to the lower or no glass particles in CoJet and control groups, respectively. On the contrary, the Rv value of the CoJet group was significantly higher than the control or FGC groups. This could reflect the effect of a deeper penetration of the CoJet sand particles compared to the FGC particles.

XRD pattern showed the presence of monoclinic phase in all zirconia specimens, including the control group, and broadened tetragonal peaks for the other groups. The monoclinic phase was found in the as-sintered zirconia (control) in previous studies [5,36,37]. Subasi et al. claimed that the presence of the monoclinic phase in control groups is a material-dependent factor [34]. IPS e.max ZirCAD is presented commercially in form of partially sintered blocks that could already contain monoclinic crystals [38]. Broadened tetragonal peaks of treated zirconia reflect the presence of rhombohedral or distorted tetragonal zirconia, in agreement with a previous study [39]. Rhombohedral zirconia is formed primarily of tetragonal and cubic crystals which could be present in partially sintered zirconia [40]. The transformability of tetragonal to monoclinic crystals depends on yttria content [41], grain size [42], sintering temperature, and holding sintering time [43]. Higher yttria content and smaller grain size provoke less tetragonal to monoclinic transformation [41,42]. In the current study, the Y-TZP used is IPS e.max ZirCAD which contains 3 mol% wt yttria, for which the grain sizes above 1µm might provoke a spontaneous transformation of t-m crystals [42]. Monoclinic crystals above 50% could adversely affect the mechanical properties of Y-TZP [44]. In the current study, the highest amount of monoclinic crystals did not exceed 27%, indicating that all the surface treatments employed in the study do not jeopardise the mechanical structure of Y-TZP. However, surface treatments which induced the lowest monoclinic crystals are still more favourable.

Shear bond strength (SBS) is a simple and reliable testing method to investigate bond strength between dissimilar materials, especially when the bonding surface area has an average diameter of 5 mm [45]. It was frequently used as a method of bonding different materials to zirconia [46,47,48]. Microtensile bond strength is a more valid method of testing the adhesion strength between two different materials compared to SBS [49]. In microtensile bond strength testing, the two adhered materials are sectioned into slices of an average of 1mm thick and mounted into a universal testing machine for the tensile testing [49]. Shear bond strength testing induces uneven distribution of stresses at the interface tested leading to cohesive failure in the substrate, and leave the interface intact, which could make the results doubtful [50]. Zirconia is very hard and difficult to be sectioned after full sintering, and cohesive failure within zirconia was never found in this study. This indicates that the shear bond strength values represented the bond strength at the interface and eliminated the doubts of the results. Furthermore, Valandro et al. reported no significant difference between microtensile versus shear bond strength testing of resin to zirconia [51]. In addition, shear bond better reflects the clinical behavior than microtensile testing [50]. For these reasons, shear bond strength was chosen as the testing method in the current study. According to ISO standards recommendations for dental materials testing, the interface between the two bonded materials should be loaded in shear at a cross speed of 0.75 ± 0.30 mm/min till debonding [52]. The choice of 0.5 mm/min as a crosshead speed still matches with ISO standards recommendations, and it was based on previous studies before [48,53,54]. 

Sandblasting zirconia with silica-coated alumina particles creates surface microretentive areas and enriches zirconia surface with silica in a process called silicatisation [55]. Silicatisation is the detachment and embedding of silica coat, off the sandblasting alumina particles, into zirconia surface [55]. Resin bonding to silica-rich zirconia surface, due to silicatisation, has been higher than to the traditional alumina blasting [15,56,57,58]. Silica coat is an essential substrate for chemical bonding to silane primer through the formation of siloxane bonds, suggesting an improved bond strength of resin composite to zirconia [55]. However, it has been reported that the detached silica coats might be loosely attached to zirconia, leaving it with insufficient substrate for the subsequent silane treatment [59]. Further, alumina particles have been found to induce microcracks on the zirconia surface which might compromise the interface between zirconia and resin [13,60,61].

Alumina-free glass-ceramics have been compared to the conventional silica-coated alumina to blast zirconia in a previous study [17]. Authors reported significantly higher bond strength of resin composite to alumina-free blasted zirconia compared to silica-coated blasting approach (24 ± 3.6 MPa and 18.7 ± 3.8 MPa, respectively). Fluorapatite glass-ceramics are alumina-free and have been used as a durable veneering material to zirconia cores in ZBR [18,62]. However, the use of fluorapatite glass-ceramic to blast zirconia has never been studied according to the published literature. Blasting zirconia with fluorapatite glass-ceramic powder in the present study created micromechanical retention on the zirconia surfaces and enriched the surfaces with silica. This could have contributed to the higher mean shear bond strength of the fluorapatite-sandblasted groups compared to other groups for the same repair materials (Table 2). This finding is consistent with the previous work of Wandser et al., who compared alumina-free and alumina-based sandblasting approaches on resin bonding to zirconia using the SBS test [17].

FGC + Nd: YAG is an alternative approach that hypothesised an effective mechanical roughening to zirconia with the potential fusion of FGC powder into the surface. The effect of this approach has been investigated and compared to other surface treatments for the first time in the current study. FGC + Nd: YAG approach reported non statistically significant lower SBS compared to FGC sandblasting regardless of the repair material (Table 2). Nd: YAG laser works by melting down ceramic substrate during a short irradiation time [15]. Shortly molten zirconia and FGC powder could have formed a hybridisation of both materials. Madani et al. used an output power of 8W in the silica-lasing method. However, in the current study, an output power of 4 W was chosen. Increasing the output power of Nd: YAG laser above 4 W has been reported to jeopardise the crystalline structure of zirconia and induce monoclinic crystals in addition to severe surface cracking [15]. However, silica-lasing was used to modify Ni–Cr alloy which lacks the sensitive crystalline structure of zirconia and higher output powers were safe. The irradiation to zirconia samples was conducted manually which might have left some areas unexposed to laser strikes, yet this has still significantly increased SBS compared to CoJet sand or control groups. The overall laser-stricken areas could have contained higher glass particles, compared to the detached silica from CoJet sand, which explains the significantly higher SBS compared to CoJet. Although the nature of the combination of FGC powders and irradiated zirconia is unknown, it seemed to be an effective approach to modify the zirconia surface for resin bonding.

CoJet sand has been a successful sandblasting approach that reported mean SBS of resin composite to zirconia ranging from 17.16 ± 1.19 MPa [63] to 22.9 ± 3.1 MPa [64]. Unless sandblasting pressure is increased, silica coats embedded in zirconia surface have been found insufficient, as confirmed by energy dispersive spectroscopy (EDS), in one study [65]. However, higher pressures might induce damaging cracks into the zirconia surface which might question the durability of repair [66]. Monobond Plus was used in the current study as it contains silane and 10-MDP monomers. Silane is a functional monomer that binds chemically to glass particles and the resin through siloxane bonds [20]. 10-MDP has two functional groups; phosphate group which binds to the hydroxyl group in zirconia and methacrylate group that copolymerises with the resin composite [1]. Monobond Plus has been used by many authors to study and investigate the bond strength of resin to composite [20,66,67,68,69].

Lithium disilicate (IPS e.max CAD) has been used as a potential repair material for zirconia in the current study. The results showed that IPS e.max CAD has reported the highest mean SBS (36.43 ± 2.3 MPa) compared to the other repair materials with statistical significance (*p* < 0.05) in all surface treatments. This finding is consistent with Yilmaz et al. [70] and the same authors in another study [71], who reported SBS (33.03 ± 5.05 MPa and 31.89 ± 5.83 MPa, respectively) of lithium disilicate to zirconia using CAD-on technique. In this technique, lithium disilicate and zirconia are manufactured separately by CAD/CAM technology and then adhered together through a low fusing glass at a temperature of 840 °C [70]. CAD-on technique has been a successful approach for adhering lithium disilicate to zirconia compared to the traditional layering in previous works [70,71,72]. However, the CAD-on technique is not applicable for zirconia repair as it necessitates the fusion of both zirconia core and lithium disilicate outside the patient mouth. In the current study, lithium disilicate was luted to zirconia using 10-MDP primer/resin cement which can be performed in the patient’s mouth. The selection of lithium disilicate as a repair material might be preferable when both mechanical and optical properties are important. IPS e.max CAD has a flexural strength of 262 ± 88 MPa and fracture toughness of 2.5 MPa.m^1/^ [73]. These mechanical properties make IPS e.max CAD a durable restorative material in anterior or posterior teeth [73].

Feldspathic porcelain (VITA VM13) has been used in the current study as a repair material, in addition to resin composite, for fractured ZBR. The results of our study showed a maximum SBS of 20.30 ± 2.22 MPa in the FGC group with no statistically significant difference (*p* > 0.05) compared to Tetric EvoCeram Bulk Fill resin composite (21.48 ± 2.06 MPa), which is in agreement with [74]. The selection of either repair material would depend on many factors, such as the aesthetic requirements, cost, and time constraints. Feldspathic porcelain is a wear-resistant, tough material and could retain its shade with no stain uptake throughout the functional time in the mouth [75,76]. It has been considered the veneering material of choice due to its optimal optical properties which match the tooth shades [77]. Furthermore, it could receive external staining which mimics the finest morphological features of natural teeth for the best outcome results [78]. However, fabrication of the feldspathic porcelain would involve a laboratory step and increase the overall cost and the time of the treatment. According to the result of the current study, the use of direct resin composite could be a more feasible option as an immediate treatment solution for fractured ZBR. Direct resin composites are easy to place and do not necessitate a laboratory step which makes the treatment faster and cheaper. However, resin composites could be stained and might need occasional polishing sessions to restore the original shade [76].

However, this in-vitro study investigated one type of zirconia substrate with the application of one primer/adhesive and employed single parameters of one laser device. Different laser devices and parameters, other types of zirconia substrates, and primers/adhesives should also be tested to provide a more concise prediction of how laser power can be an effective alternative approach in zirconia repair protocols.

## 5. Conclusions

It is concluded that t-m phase transformation depends on the surface treatment employed, where blasted zirconia with CoJet sand or Nd: YAG laser treatment to FGC-coated zirconia revealed the highest number of monoclinic crystals compared to FGC-blasted zirconia. Laser-fused FGC showed the highest surface roughness to zirconia compared to FGC or CoJet. FGC-blasted zirconia reported a higher amount of silica compared to CoJet sand and contributed to a significant increase in the shear bond strength of either feldspathic porcelain or lithium disilicate to zirconia. Sandblasting zirconia with FGC powders and FGC + Nd: YAG could be effective alternative approaches for zirconia repair. While direct resin composite and feldspathic porcelain could work as repair materials for fractured ZBR, lithium disilicate might be a better choice especially when high mechanical properties are a prerequisite.

## Figures and Tables

**Figure 1 materials-14-01468-f001:**
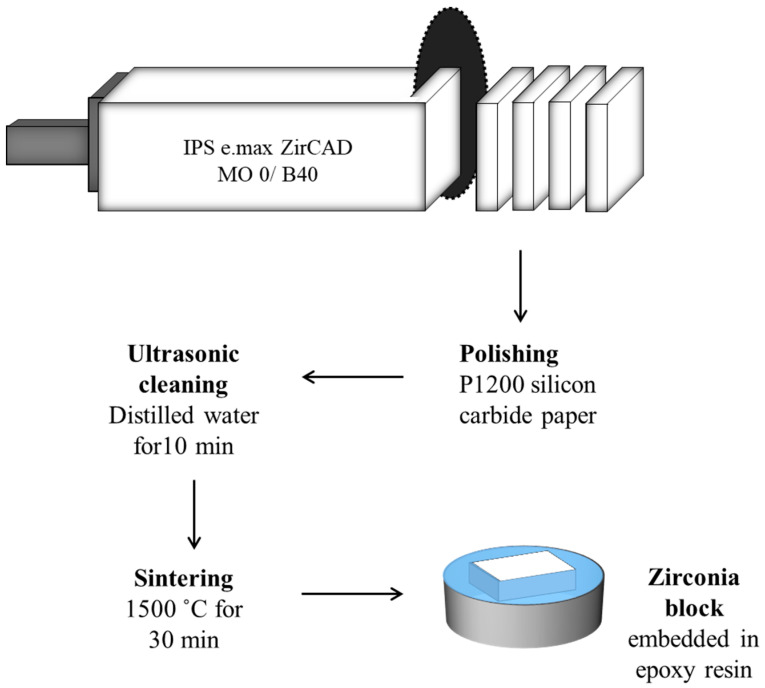
Schematic diagram showing the flow of steps of zirconia specimens’ preparation.

**Figure 2 materials-14-01468-f002:**
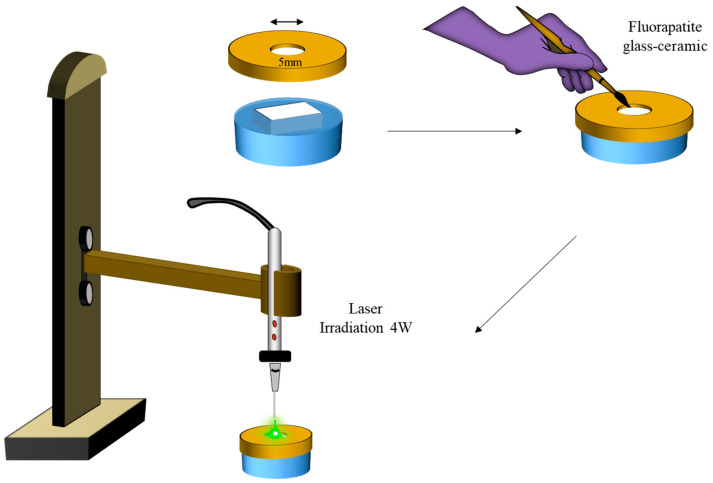
Schematic diagram showing FGC+ Nd: YAG approach. A brass metal cover with a central hole of 5 mm in diameter and 0.5 mm thickness was used to define the area of the zirconia surface to be irradiated with a laser. A paint brush was used to coat the exposed area with FGC powder. Nd: YAG laser handpiece was mounted into a carrier so that the irradiation was perpendicular to the exposed zirconia surface at a fixed distance of 1 mm.

**Figure 3 materials-14-01468-f003:**
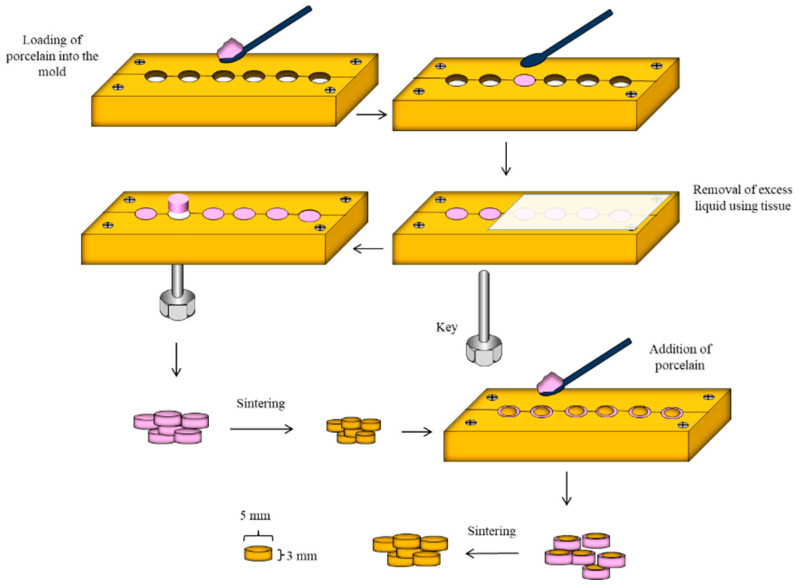
Schematic diagram showing detailed steps of VITAVM^TM^ 13 porcelain specimens’ preparation.

**Figure 4 materials-14-01468-f004:**
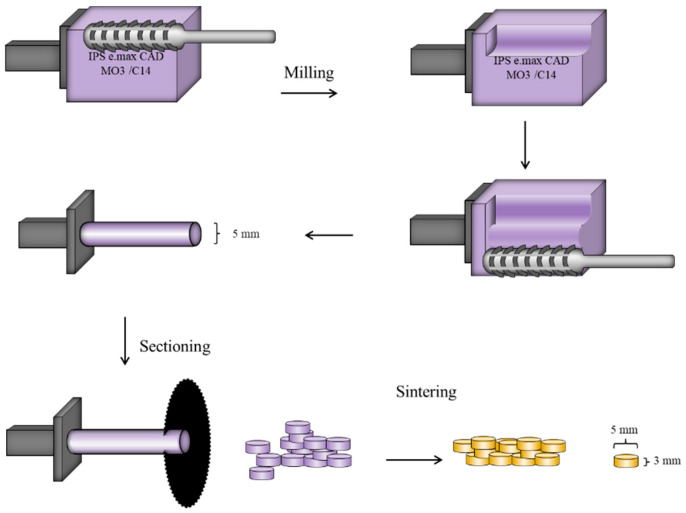
Schematic diagram showing the detailed steps of preparation of IPS e.max CAD lithium disilicate specimens.

**Figure 5 materials-14-01468-f005:**
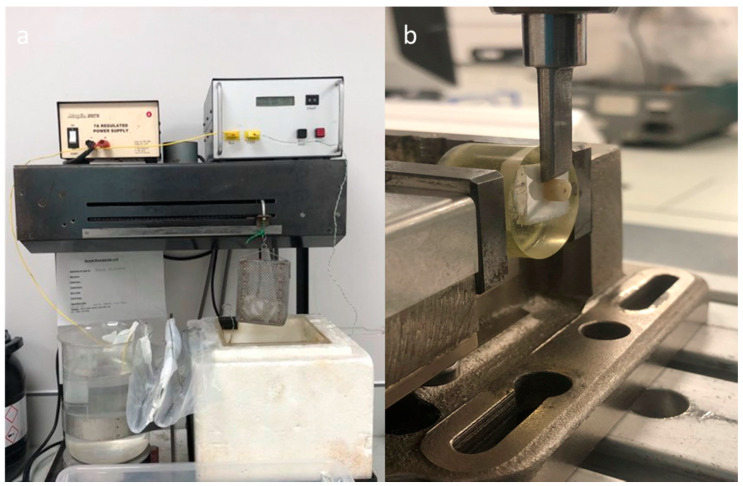
Bonded specimens underwent 10,000 thermocycles between 5 °C and 55 °C (**a**) before being tested for shear bond strength (SBS) (**b**).

**Figure 6 materials-14-01468-f006:**
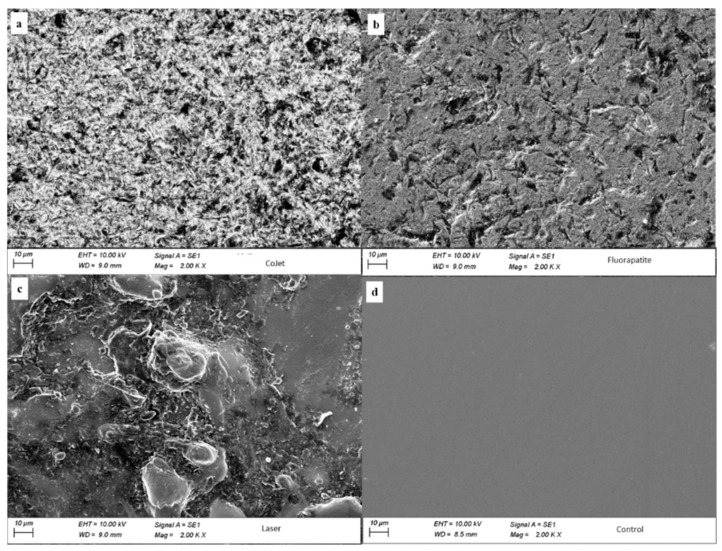
Scanning electron microscope images 2000× showing an extensive irregular surface of the CoJet group (**a**), less irregular surface with scattered grooves in FGC group (**b**), blister-like deformities in FGC + Nd: YAG (**c**), and plain smooth surface in the control group (**d**).

**Figure 7 materials-14-01468-f007:**
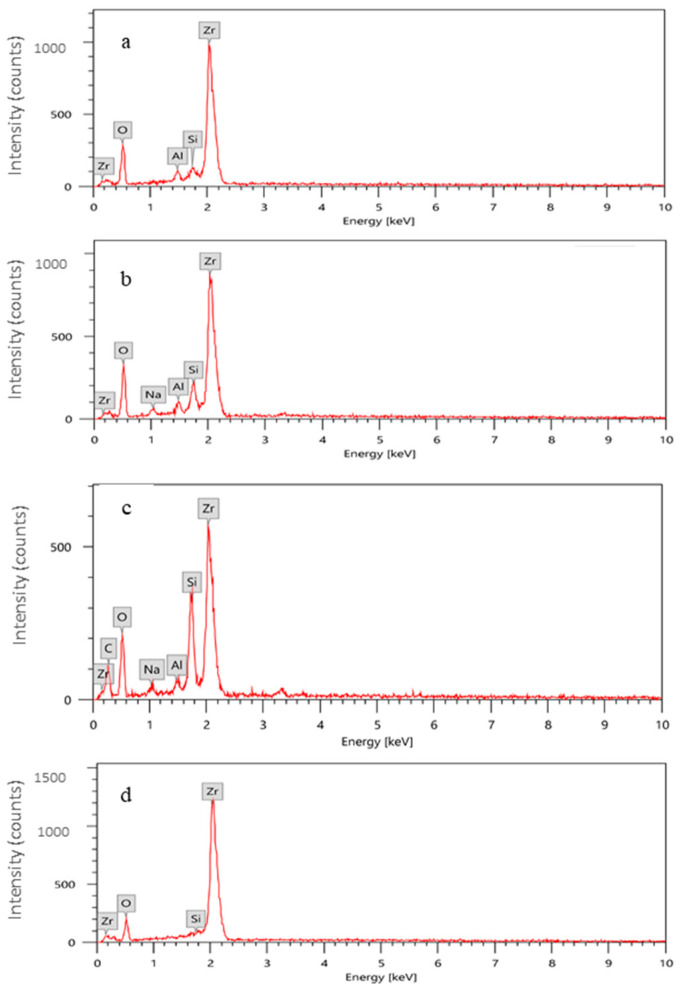
Energy dispersive spectroscopy (EDS) analysis showing the percentage mass of silica in groups (**a**) CoJet (1.25 ± 0.13%), (**b**) FGC (3.42 ± 0.17%), (**c**) FGC+Nd: YAG (6.82 ± 0.21%), and (**d**) Control, where silica content was not detected.

**Figure 8 materials-14-01468-f008:**
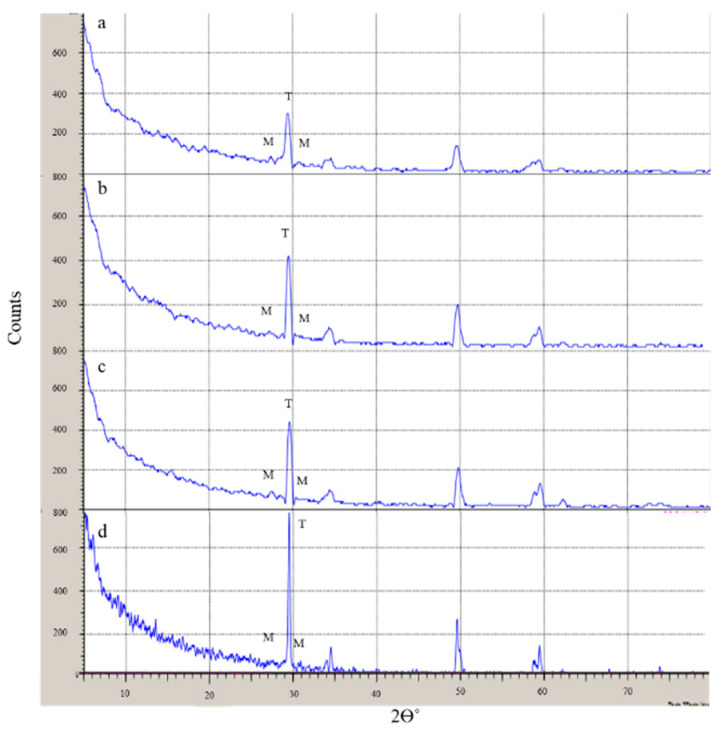
XRD diffraction pattern for all studied groups; (**a**) CoJet, (**b**) FGC, (**c**) FGC+ Nd: Laser, and (**d**) Control. Relative intensities of tetragonal and monoclinic peaks were marked at I_T_ (101) and I_M_ (111 and 111^−^) by letters T and M, respectively.

**Figure 9 materials-14-01468-f009:**
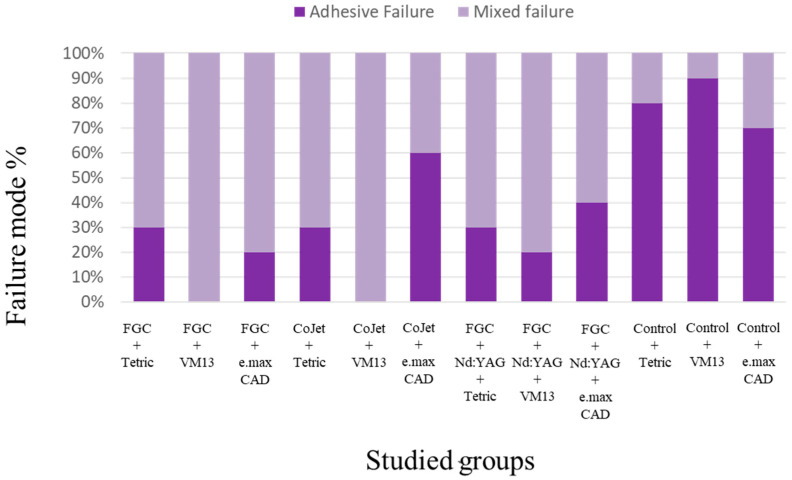
Bar chart showing the failure mode% in all 12 studied groups. The deep purple colour represents the percentage of the adhesive mode of failure in the debonded specimens, while the light purple colour represents the mixed adhesive/cohesive failure.

**Figure 10 materials-14-01468-f010:**
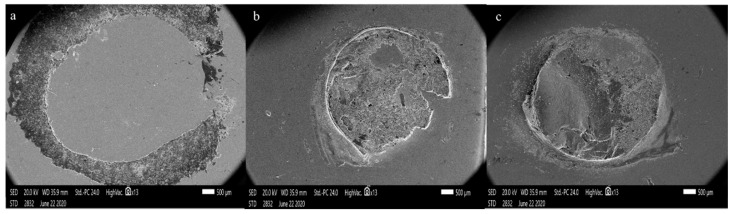
SEM images of different modes of failure of debonded specimens X13; (**a**) adhesive failure in Group D, (**b**) mixed adhesive cohesive failure in a Group A, and (**c**) is a mixed adhesive cohesive failure with a cohesive fracture in Group B.

**Table 1 materials-14-01468-t001:** Descriptive statistics and Tukey’s post hoc results of the mean surface roughness values arithmetic average of surface profile (Ra), an average of maximum peaks (Rp), the average of deepest valleys (Rv), and the maximum average of the highest peak and lowest valleys (Rz) (µm) of groups A (CoJet), B (FGC), C (FGC + Nd: YAG), and D (Control).

				Mean (SD) µm				
Groups	Ra	95% CI	Rp	95% CI	Rv	95% CI	Rz	95% CI
Group A (CoJet)	0.29 (0.02) ^A^	0.27–0.3	0.66 (0.07) ^A^	0.59–0.72	0.92 (0.05) ^B^	0.87–0.96	2.75 (0.45) ^B^	2.33–3.17
Group B (FGC)	0.31 (0.05) ^A^	0.27–0.36	1.08 (0.12) ^B^	0.97–1.19	0.51 (0.11) ^A^	0.41–0.61	3.28 (0.37) ^B^	2.93–3.62
Group C (FGC + Nd: YAG)	2.41 (0.38) ^B^	2.06–2.77	5.66 (0.6) ^C^	5.09–6.21	3.58 (0.25) ^C^	3.35–3.82	14.82 (1.85) ^C^	13.11–16.53
Group D (Control)	0.03 (0.01) ^A^	0.01–0.04	0.24 (0.01) ^A^	0.23–0.25	0.30 (0.02) ^A^	0.28–0.32	0.36 (0.09) ^A^	0.28–0.44

No statistical significance (*p* > 0.05) is indicated by the same superscript capital letter in columns when comparing different surface treatments.

**Table 2 materials-14-01468-t002:** Descriptive statistics and Tukey post hoc test of the mean shear bond strength in MPa (SD) and 95% confidence interval (CI) bound of different repair materials to zirconia according to different surface treatments.

			Mean SBS MPa (SD)					
Groups	CoJet	95% CI	FGC	95% CI	FGC + Nd: YAG	95% CI	Control	95% CI
Tetric EvoCeram Bulk Fill	19.3 (0.8) ^Aa^	18.72–19.91	21.5 (2.1) ^Aa^	20–22.95	19.1 (1.2) ^Aa^	18.19–19.97	15.5 (1.5) ^Ab^	14.42–16.56
VITA VM 13	14.2 (2.1) ^Bb^	12.69–15.66	20.3 (2.2) ^Aa^	18.72–21.89	20.3 (1.9) ^Aa^	18.92–21.58	13.3 (2.1) ^Ab^	11.76–14.8
IPS e.max CAD	25.8 (2.40) ^Cb^	24.11–27.56	36.4 (2.3) ^Ba^	34.76–38.09	35 (2.2) ^Ba^	33.39–36.5	22.7 (1.5) ^Bc^	20.63–23.26

No statistical significance (*p* > 0.05) is indicated by the same superscript capital letter in columns when comparing different repair materials for the same surface treatment, and by same superscript small letters when comparing the different surface treatments for the same repair materials.

## Data Availability

The data are contained in the article.

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
