# Peer review of "Effect of Air-Abraded Versus Laser-Fused Fluorapatite Glass-Ceramics on Shear Bond Strength of Repair Materials to Zirconia"

_materials, 2021, doi:10.3390/ma14061468_

Round 1
Reviewer 1 Report
This manuscript is of a low sientific level.
This manuscript cannot be published. I think the best way is to reject it.
There are literatures that describe the effect of airborne Cojet particles on the surface or zirconia very well. It is not reccomended to use these particles.
Fluor appatit particles cannot roughen the surface of zirconia and I
don´t believe that this material has a future as airborne material. Also, I do not think that it can be used as a powder for laser treatment.
Reviewer 2 Report
Dear authors
Something went wrong while submitting the manuscript. Pay attention to numerous places in which the text is: Error! Reference source not found (lines 135, 189, 206, 268, 274, 289, 318, 344, 430, 438). Please correct.
The metodology is excellent and so is the documentation of the results. I have several recommendation before accepting the manuscript:
line 38: In case of a fractured ZBR, the treatment of choice is the replacement of the prosthesis [7]. This statement is incorrect. Fortunately in most cases the ZBR can be adjusted and polished, without the need to replace (J Adv Prosthodont 2012, 4: 76-83). I recommend to adapt the criteria made by Heintze (grades 1,2,3). Accordingly the frequency of grades 1 and 2 are considerably higher than grade 3 (Int J Prosthodont 2010, 23:493-502). Please rephrase and correct.
line 219- Add to the Tetric subgroup also: monobond plus.
line 271- Add in brackets after Group C (Figure 5d).
Figure 5 is confusing. Match the order of the images to the groups; Group D will match Figure 5d, Group A will match Figure 5a and so on. Correct the text accordingly.
Strangely there is two times Figures 2, 3 and 4. Change accordingly to Figures 7, 8 and 9.
line 354- Figure 3 which is actually Figure 8 should have in the legend the meaning of the different colours.
Add to the results some description of Figure 4 which is Figure 9 (SEM images of failures).
Reference 26 relate to fixed partial dentures and not crowns. In fixed partial dentures the rate of fractures is higher compared to crowns. Please elaborate on that issue. In crowns the rate is much lower (see: Monaco Carlo et al 2013). Also when referring to fixed dental prosthesis the rate in more recent publications is better than the one cited (Teichmann M, Clinical Oral Investigations 2018).
Reviewer 3 Report
Effect of air-abraded versus laser-fused fluorapatite glass-ceramics on shear bond strength of repair materials to zirconia
MANUSCRIPT NUMBER: materials-1122030
The aim of the present investigation was to evaluate the effect of sandblasting with FGC powders or tribochemical silica-coated alumins and FGC+ND:YAG approaches on shear bond
strength (SBS) of different repair materials (feldspathic and lithium disilicate glass-ceramic, and resin composite) to zirconia.
GENERAL COMMENTS
The study is original and interesting. The investigation methodologies are appropriate for the paper topic. The results are well-presented and the study evidence were clearly discussed with recent and appropriate bibliography. The paper is recommended for publication after a minor revision.
Introduction
The present section is clear and fluent. The paragraph could be improved introducing the novel application of Nd:YAG laser to improve the thickness of TiO2 and for photobiomodulation procedures:
Scarano A, Lorusso F, Postiglione F, Mastrangelo F, Petrini M. Photobiomodulation Enhances the Healing of Postextraction Alveolar Sockets: A Randomized Clinical Trial With Histomorphometric Analysis and Immunohistochemistry. J Oral Maxillofac Surg. 2021 Jan;79(1):57.e1-57.e12. doi: 10.1016/j.joms.2020.09.008. Epub 2020 Sep 17. PMID: 33058773.
Several references are wrong: “Error! Reference source not found”
Please correct the citations in the main text
Methods
The authors should include in the manuscript more photographs of the study samples and the thermocycles loading experiment.
Results
The A, B, C and D groups treatment should be described in the caption of tab. 1. The 95% confident intervals could be added in the descriptive statistics. The leverl of significance of groups comparison is not clearly presented. I suggest to add a new table for the group comparison and the multivariate statistical analysis output.
Discussion
The shear loading set up should be discussed in this section. A loading of 500N seems to be more appropriated as occlusal loading force to perform a static axial load. The authors should discuss the rationale of this choice.
Round 2
Reviewer 1 Report
There are literatures on Cojet and other treatment methods detailing the effect of sanblasting.
DOI:10.1016/j.dental.2016.02.001
DOI:10.1016/s0109-5641(99)00070-6
DOI:10.1016/j.surfcoat.2012.04.043
After reading these papers, you can better understand the effect Cojet and other treatments.
Please improve Fig. 9 and conclusions
